# Position: Interpretability in Deep Time Series Models Demands Semantic Alignment

**Giovanni De Felice**[* 1]  **Riccardo D'Elia**[* 2 3]  **Alberto Termine**[2 3]  **Pietro Barbiero**[4]

**Giuseppe Marra**[5]  **Silvia Santini**[1]

## Abstract

Deep time series models continue to improve predictive performance, yet their deployment remains limited by their black-box nature. In response, existing interpretability approaches in the field keep focusing on explaining the internal model computations, without addressing whether they align or not with how a human would reason about the studied phenomenon. Instead, we state **interpretability in deep time series models should pursue *semantic alignment***: predictions should be expressed in terms of variables that are meaningful to the end user, mediated by spatial and temporal mechanisms that admit user-dependent constraints. In this paper, we formalize this requirement and state that, once established, semantic alignment must be preserved under temporal evolution: a constraint with no analog in static settings. Provided with this definition, we outline a blueprint for semantically aligned deep time series models, identify properties that support trust, and discuss implications for model design.

## 1. Introduction

Deep learning (DL) has transformed time series analysis and now dominates empirical benchmarks (Benidis et al., 2022; Wang et al., 2024). Yet while DL models produce accurate predictions, the *how* and *why* a prediction was made remains difficult for humans to scrutinize, due to the inherent *opacity* of such models (Burrell, 2016; Facchini & Termine, 2021).

This lack of *interpretability* poses significant challenges

for the wider adoption of DL models in several critical scenarios, including finance (Arsenault et al., 2025) and healthcare (Di Martino & Delmastro, 2023), due to safety-related risks, ethical considerations, and the need to comply with regulatory frameworks (Goodman & Flaxman, 2017; Act, 2024). Additionally, the reliance on non-interpretable "black-box models" for scientific discovery "*can decrease user trust in predictions and have limited applicability in areas where model outputs must be understood before real-world implementation*"(Wang et al., 2023).

To cope with these challenges, vast numbers of approaches to enable interpretability (and/or explainability) of DL models have been developed (Turbé et al., 2023; Zhao et al., 2023; Delaney et al., 2021; Arsenault et al., 2025). Methods explicitly targeting time series data include post-hoc explanations (Tonekaboni et al., 2020; Bento et al., 2021; Zhao et al., 2025), attention-based models (Lim et al., 2021), linearized dynamics (Kumar et al., 2024), prototype-based methods (Huang et al., 2025). While effective for inspecting model behavior, these approaches largely focus on exposing internal computations or sensitivities, without addressing whether they align or not with how domain experts would reason about the studied phenomenon. Recently, mechanistic interpretability methods (Pandey et al., 2025; Wiliński et al., 2025; Kalnāre et al., 2025) have begun to observe correspondences between internal representations and human concepts, yet such alignment remains incidental rather than structural.

We argue that this misalignment is fundamental and remains a primary limiting factor towards wider, safer, and more transparent deployment of deep time series models. We thus posit that **deep time series models should enforce *semantic alignment* (SA) by making the variables and internal inference mechanisms used by the model correspond to those through which domain experts reason about the phenomenon**. Clinicians may struggle to reason in terms of "timestep 47", but can relate to constructs such as "onset of tachycardia"; an engineer may immediately relate to "thermal stress accumulation", but be unable to do so with "hidden unit activations". When a model's internal variables and dynamics are not structurally enforced to align with

---

[*]Equal contribution  [1]Università della Svizzera Italiana (USI, CH) [2]University of Applied Sciences and Arts of Southern Switzerland (SUPSI, CH) [3]Dalle Molle Institute for Artificial Intelligence (IDSIA, CH) [4]IBM Research (CH) [5]KU Leuven (BE). Correspondence to: Giovanni De Felice <giovanni.de.felice@usi.ch>.

*Proceedings of the $43^{rd}$ International Conference on Machine Learning*, Seoul, South Korea. PMLR 306, 2026. Copyright 2026 by the author(s).

domain-level concepts, users cannot meaningfully validate or intervene on its behavior.

This issue has been recognized in the broader scope of machine learning (Rudin, 2019; Freiesleben & König, 2023), calling for models that are interpretable *by design* and align with human-understandable attributes or abstractions. While this call has begun to be addressed in static domains, such as in computer vision, in particular through concept-based and neurosymbolic approaches (Kim et al., 2018; Poeta et al., 2023; Bhuyan et al., 2024), it remains unanswered in the time series community. Importantly, the field lacks a mathematical formulation of SA that can be operationalized and enable the development of effective methods to achieve interpretable deep time series models.

To spur further research in this direction, we ground our position in a **formal definition of SA for time series models** (Sec. 3), and later propose a blueprint to guide the future development of **interpretable deep time series models** (Sec. 4). Crucially, we discuss **properties** that emerge from the blueprint and support model trustworthiness, and we speculate on new **model design opportunities** (Sec. 5). Finally, we examine **alternative views** to support interpretability within (and beyond) the time series domain, and highlight that SA can be used to achieve interpretability without compromising accuracy (Sec. 6).

# 2. Deep Learning for Time Series

This section contextualizes the paper within the literature of DL for time series by formalizing the problem setting, our underlying assumptions, and the class of considered models.

## 2.1. Problem setting

**Data.** We consider a time series

$$\mathbf{x}_{\leq T} := (x_0, x_1, \ldots, x_T)$$

given by realizations of a stationary stochastic process $\{X_t\}_{t \in \mathcal{T}}$ taking values in $\mathcal{X}$, with $\mathcal{T} = \{0, \ldots, T\}$. For ease of exposition, we focus on the case of a single input sequence; the discussion can be easily extended to include exogenous inputs.

**Predictive task.** We consider a broad class of predictive tasks on time series, unified under a common probabilistic formulation. The learning problem is to model the conditional distribution

$$P(Y \mid \mathbf{x}_{\leq t}),$$

where $Y$ is a task-specific output variable taking values in $\mathcal{Y}$. The output space $\mathcal{Y}$ instantiates differently depending on the task: *(i)* $\mathcal{Y} = \mathcal{X}^L$, where $Y$ is a window of length $L$, covering, e.g., forecasting, sequence generation, denoising,

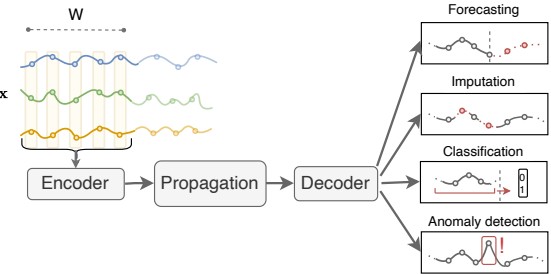

*Figure 1.* Template architecture for time series tasks, characterizing the class of models considered in this work. Observations within a window are first mapped by an *Encoder* into latent representations, which are transformed by a task-specific *Propagation* module and subsequently mapped by a *Decoder* to the target output. Modules operating on latent representations are highlighted in gray.

or imputation; *(ii)* $\mathcal{Y} = \{1, \ldots, K\}$, and $Y$ denotes a discrete variable, as in, e.g., classification or event detection; *(iii)* $\mathcal{Y} = \mathbb{R}$, and $Y$ is a scalar variable, as in, e.g., regression tasks.

## 2.2. Considered class of models

We consider a generic class of models, unified under an *encoding–propagation–decoding* template (Fig. 1). At each time $t$, the model class is defined as

$$\begin{aligned} \textbf{Encoding:} \quad & \mathbf{u}_t = \text{ENC}(\mathbf{x}_{\leq t}), & \text{(1a)} \\ \textbf{Propagation:} \quad & \mathbf{z}_{t+1} = \text{PROP}(\mathbf{z}_{\leq t}, \mathbf{u}_t), & \text{(1b)} \\ \textbf{Decoding:} \quad & \hat{\mathbf{y}} = \text{DEC}(\mathbf{z}_{t+1}). & \text{(1c)} \end{aligned}$$

where $\mathbf{u}_t \in \mathcal{U}$ and $\mathbf{z}_t \in \mathcal{Z}$ denote learned representations produced by each parametric module, which can be modeled as realizations of stochastic processes $\{U_t\}_{t \in \mathcal{T}}$ and $\{Z_t\}_{t \in \mathcal{T}}$, of which we assume the existence. We denote with $\hat{\mathbf{y}} \in \mathcal{Y}$ output predictions by the model.

This framework generalizes classical state-space representations, of which recent results have highlighted the generality (Hauser et al., 2019; Gu et al., 2022; Muca Cirone et al., 2024). While other formulations are possible, the one above provides a common template for a broad class of time series models, including recurrent neural networks (Hochreiter & Schmidhuber, 1997; Cho et al., 2014), temporal convolutional networks (LeCun & Bengio, 1998; Bai et al., 2018), state-space models (Gu & Dao, 2024), attention-based architectures (Wu et al., 2021a), and spatio-temporal graph neural networks (Corradini et al., 2025; Cini et al., 2025). A concrete instantiation is provided in App. B.1.

**Observation 2.1** (Extensions)**.** The formulation above extends naturally to collections of multiple time series (e.g., sensor networks), where the propagation layer exchanges information across series via a graph structure (Jin et al., 2024a), and to discrete token sequences (e.g., text), as in natural language processing (Mariet & Kuznetsov, 2019).

### 2.3. The opacity problem

In DL literature, *opacity* refers to the problem of understanding *how* and *why* a model produces a certain output. We distinguish two forms: *structural* (or "access") and *semantic* opacity (Facchini & Termine, 2021; Boge, 2022).

**Structural opacity.** Structural opacity concerns understanding the mathematical and computational structure of a model (the "how"). In time series models, this includes understanding the role of recurrent states, attention weights, message-passing steps, or convolutional filters. This is the form of opacity most commonly addressed in the mechanistic interpretability community (Bereska & Gavves, 2024).

**Semantic opacity.** A distinct and arguably more consequential form of opacity concerns the semantics of the representations involved in the model reasoning. We refer to this as *semantic opacity*. Informally, a model is semantically opaque when its computational steps cannot be expressed in terms of meaningful variables and inferential structures used by domain experts. Importantly, semantic opacity is orthogonal to structural opacity: a model may be easy to explain in terms of its internal computations (structural transparency) but still be difficult for humans to relate them to meaningful concepts for the task at hand (semantic opacity).

To help guide the exposition, we adopt a simple running example drawn from industrial monitoring.

*Running Example. Consider an engineer monitoring an industrial system where a deep model predicts equipment failure from temperature sequences* $\mathbf{x}_{\leq t}$. *When the model predicts imminent failure, the hidden states,* $\mathbf{u}_t$ *and* $\mathbf{z}_t$ *are irrelevant for operational decision-making. The engineer, by contrast, reasons in terms of* overheating *(temperature exceeding a critical threshold),* thermal stress *(cumulative exposure to high temperatures), and* degradation *(stress-induced component wear). The mismatch between these domain concepts and the model's internal representations gives rise to semantic opacity.*

Given this definition, we observe that representations $\mathbf{u}_t$ and $\mathbf{z}_t$ in the considered class of models (Eq. 1a-1b) are semantically opaque in general. In the case of a supervised task, the only component capable of reassigning semantics is the decoder, as the output variable $\hat{\mathbf{y}}$ is typically aligned with a human-provided label.

## 3. Semantic Alignment for Time Series

In the following, we formalize the notion of SA as the response to semantic opacity.

### 3.1. Notation and terminology

**Concepts.** For modeling purposes, we adopt a probabilistic perspective and assume human reasoning can be formalized as if it operates over *random variables* (Chater et al., 2006; Tenenbaum et al., 2011). When a random variable of this kind carries an interpretation meaningful to humans, we refer to it as a *concept* (Kim et al., 2018) [1], denoted by

$$C : \Omega \to \mathcal{C}.$$

Concept variables can range over a variety of domains, including binary labels ($\mathcal{C} = \{0, 1\}$), categorical variables ($\mathcal{C} = \{1, \ldots, K\}$), real-valued quantities ($\mathcal{C} = \mathbb{R}$), or other structured spaces. This modeling choice leads naturally to considering the temporal extension of concepts as stochastic processes $\{C_t\}_{t \in \mathcal{T}}$.

**Mechanisms.** Reasoning also involves understanding how variables depend on one another and how they evolve over time. We term *mechanisms* the relations between concepts, and model them as *conditional probability distributions* (CPDs) $P(V_{\text{out}} \mid V_{\text{in}})$, understood as the conditional law of $V_{\text{out}}$ given $V_{\text{in}}$. Equivalently, for each realization $v$ of $V_{\text{in}}$, the mechanism specifies the distribution $P(V_{\text{out}} \mid v)$ over possible values of $V_{\text{out}}$. Mechanisms can be operationalized by making explicit their parametric dependence:

$$P(V_{\text{out}} \mid V_{\text{in}}; \boldsymbol{\theta}).$$

There is flexibility in how the parametrization is realized in practice. A standard approach would be to parameterize CPDs with neural network weights, which model the distribution (and hence the dependence) implicitly. An arguably more interpretable setup is to assume that the family of a CPD is known and use a neural network to predict its parameters given the parent concepts. A practical example would be a layer predicting the activation probabilities of a categorical distribution, or, for continuous concepts, a layer predicting the mean and variance of a Normal distribution: $\mathcal{N}\big(\mu = \text{NN}_1(v), \sigma^2 = \text{NN}_2(v)\big)$ [2].

### 3.2. Semantic alignment

SA consists of matching the model's internal variables and mechanisms with the corresponding entities that are understandable to the intended domain-expert. In this section, we begin with the alignment of model variables and later address the alignment of mechanisms.

**Semantic alignment of model variables.** In dynamic settings, we can distinguish two types of concepts that an

---

[1]More formal definitions arise in formal concept analysis (Ganter et al., 1999), where representational variables are characterized by grouping entities that share a set of properties.

[2]Note how deterministic mechanisms $V_{\text{out}} = \text{NN}(v)$ are recovered as the special case $P(V_{\text{out}} \mid v) = \delta_{\text{NN}(v)}$, where $\delta_a$ is the Dirac measure at $a$.

expert may be interested in modeling (Yingzhen & Mandt, 2018), each requiring distinct considerations.

- **Instantaneous concepts** $C_t^U$: concepts that represent properties of the system up to time $t$, i.e., act as "snapshots". Their semantics are understood by the human, but their temporal evolution (at $t+1$) is either not well-defined, not predictable given the observed sequence $\mathbf{x}_{\leq t}$, or not relevant for the task.

- **Dynamic concepts** $C_t^Z$: concepts whose semantics are understood by the human and whose temporal evolution is both inferable from $\mathbf{x}_{\leq t}$ and explicitly of interest to model, i.e., the user seeks to predict their future values while preserving their semantics over time.

*Running Example.* Overheating *could be an instantaneous concept representing the current temperature exceeding a critical threshold.* Thermal stress *could be a dynamic concept reflecting the cumulative exposure to high temperatures over time, which the engineer may want to forecast to prevent failures.*

The model class introduced in Sec. 2.2 is well suited to represent both types of concepts: representing instantaneous concepts via $U_t$ and dynamic concepts with $Z_t$.

Ideally, perfect alignment between two concepts $A_t$ and $B_t$ would require them to be *indistinguishable*, i.e., $P(A_t = B_t \text{ for all } t \in \mathcal{T}) = 1$. In practice, verifying this is generally intractable; we can only observe a concept conditional on a finite sample of realized trajectories. For this reason, we use an *operational* notion of alignment that requires equality of $A_t$ and $B_t$ conditional on a realized trajectory $\mathbf{x}_{\leq t}$, i.e. $P(A_t = B_t \mid \mathbf{x}_{\leq t}) = 1$ for all $t \in \mathcal{T}$.

SA in temporal domains therefore requires two conditions (Fig. 2): (i) the encoder must map raw observations to representations $U_t$ that align with instantaneous concepts $C_t^U$, and (ii) the propagation mechanisms must produce representations $Z_{t+1}$ that align with the dynamic concept $C_{t+1}^Z$ predicted from $\mathbf{x}_{\leq t}$.

*Definition* 1 (**Semantic alignment of concepts**). Consider a model as in Section 2.2 with stochastic processes $X_t$, $U_t$, and $Z_t$. Let $C_t^U$ and $C_{t+1}^Z$ be $\sigma(X_{\leq t})$-measurable concept processes. We say that $U_t$ and $Z_t$ are *semantically aligned* with $C_t^U$ and $C_t^Z$, respectively, conditional on $\mathbf{x}_{\leq T}$ if, for every $t \in \mathcal{T}$:

(i) *Sem. alignment of instantaneous concepts:*

$$P(U_t = C_t^U \mid \mathbf{x}_{\leq t}) = 1. \qquad (2)$$

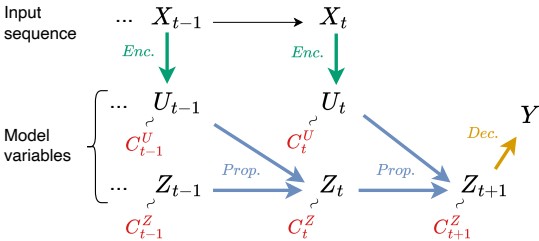

*Figure 2.* Computational graph showing the inference pathway for a model from the class described in Sec. 2.2. The symbol $\sim$ indicates SA of representations with concepts.

(ii) *Sem. alignment of dynamic concepts:*

$$P(Z_{t+1} = C_{t+1}^Z \mid \mathbf{x}_{\leq t}) = 1. \qquad (3)$$

The second requirement arises only in dynamic settings with no analog in static domains. Removing this requirement could lead the SA to decay exponentially over time.

**Observation 3.1.** Note that, in the above definition, we implicitly assume the semantics of human concepts remain consistent over time. This is not a limitation, but rather it enables the human to reliably intervene on deployed models, knowing that their understanding of the concept will not change. Relaxing this assumption could open interesting research directions.

**Observation 3.2.** When the representation and concept domains differ ($\mathcal{Z} \neq \mathcal{C}$), alignment could be defined via a *translation* function $\tau : \mathcal{Z} \to \mathcal{C}$, and the equalities above are replaced by $\tau(Z_t) = C_t$. This accommodates settings where concepts are coarser than representations. For clarity of exposition, we assume $\mathcal{Z} = \mathcal{C}$.

**Mechanism alignment as constraint satisfaction.** SA of concepts is insufficient to ensure fully interpretable models: models may represent the right concepts but relate them in ways that are too complex and thus remain opaque to human understanding. Importantly, human knowledge about mechanisms is often partial and typically specifies only *constraints*. Such constraints range from complete specification (e.g., known physical laws, or fixed functional bases) to qualitative properties (e.g., monotonicity, stability, sparsity, invariances). In practice, humans can specify the set of admissible CPDs $\mathcal{M}^{(h)}$ for a concept. This can be done by restricting $\mathcal{M}^{(h)}$, or by specifying constraints on their parametrization (e.g., restricting to monotonic or linear layers). Different users will therefore induce different $\mathcal{M}^{(h)}$, depending on their domain knowledge. In this view, we define SA of mechanisms as a constraint satisfaction problem, orthogonal to concept alignment.

*Definition* 2 (**Semantic alignment of mechanisms**). Let $P(V_{\text{out}} \mid V_{\text{in}})$ be a mechanism relating variables $V_{\text{out}}$ and $V_{\text{in}}$, and let $\mathcal{M}_{V_{\text{out}}|V_{\text{in}}}^{(h)}$ denote the set of distributions admissible under human constraints. The mechanism is *aligned* if

$$P(V_{\text{out}} \mid V_{\text{in}}) \in \mathcal{M}_{V_{\text{out}}|V_{\text{in}}}^{(h)}.$$

*Running Example. Consider a model that accurately represents temperature, thermal stress, and degradation as separate concepts. If these variables are related through a dense neural network, the user can inspect concept values, but not how they have influenced one another. Vice versa, a linear system such as $\mathbf{z}_{t+1} = \mathbf{A}\mathbf{z}_t$ may satisfy known physical constraints (e.g., monotonicity) yet operate on embeddings that are unrelated to domain concepts.*

We argue the definitions above provide a *necessary* condition for semantic interpretability in temporal settings. Developing a comprehensive theory of interpretability lies beyond the scope of this work and has only recently been explored in early efforts (Giannini et al., 2024; Tull et al., 2024; Barbiero et al., 2025). Nevertheless, addressing the temporal dimension of alignment remains underexplored even in such existing theoretical frameworks.

# 4. A Blueprint for Interpretable Deep Time Series Models

In this section, we outline a blueprint for designing semantically aligned deep time series models. We rethink how representations, mechanisms, and training objectives are structured within the standard template of Sec. 2.2.

## 4.1. Background: concept-based models

Concept-based models (CBMs) provide a foundation for enforcing SA in static prediction tasks. In their canonical form (Koh et al., 2020), concept-based models decompose prediction into two stages [3]: (i) an encoder maps inputs to a set of human-interpretable concept variables, and (ii) a predictor outputs the task variable as a function of these concepts. Formally, for an input $x$ and concept vector $\mathbf{c}$,

$$P(Y, \mathbf{c} \mid x) = P(Y \mid \mathbf{c}) \cdot P(\mathbf{c} \mid x).$$

This factorization follows from the chain rule under the design choice of a *conditional independence assumption* $Y \perp X \mid \mathbf{c}$, which forces all task-relevant information to flow through the concept bottleneck, so that predictions are expressed explicitly in terms of the specified concepts.

Traditionally, CBMs assume that concepts are conditionally independent given the input $x$, i.e.

---

[3]Here, we align our notation with the standard in the concept-based community by denoting all internal model variables as $c$.

$P(\mathbf{c} \mid x) = \prod_k P(c^{(k)} \mid x)$. Recent extensions (Dominici et al., 2025; De Felice et al., 2025) relax this assumption by modeling causal dependencies among concepts via a directed graph. Here, concepts $c^{(k)}$ are distinguished as *source* concepts, $k \in \mathcal{S}$, which are encoded directly from the input $x$, and *derived* concepts, $k \in \mathcal{D}$, which are predicted from their parent concepts $\text{Pa}(c^{(k)})$ in the graph. Specifically,

$$P(\mathbf{c} \mid x) = \prod_{k \in \mathcal{D}} P(c^{(k)}|\text{Pa}(c^{(k)})) \prod_{k' \in \mathcal{S}} P(c^{(k')}|x).$$

## 4.2. Semantically aligned deep time series models

We argue that SA in time series settings requires a *temporal extension* of the concept-based paradigm. In particular, models should introduce a *structured concept space* in which concepts are propagated by aligned mechanisms both across space and **across time**. Specifically, we hypothesize predictions could decompose as follows (concrete instantiations are provided in Fig. 3 and App. B.2):

i) **Concept encoding.** A concept encoder maps a window of raw observations (and optional exogenous inputs) to a set of source concepts

$$\prod_{k \in \mathcal{S}} P(c_{\leq t}^{(k)} \mid \mathbf{x}_{\leq t}).$$

ii) **Concept propagation.** The propagation step is composed of mechanisms that compute derived concepts by evolving concepts through time. Propagation mechanisms can be categorized as:

- *Temporal mechanisms*: $P(c_{t+1}^{(k)} \mid c_{\leq t}^{(k)})$, modeling how individual concepts evolve over time;

- *Spatio-temporal mechanisms*: $P(c_{t+1}^{(k)} \mid c_{\leq t}^{(j)}, \ldots)$, modeling temporal dependencies between different concepts.

iii) **Task decoding (optional).** A decoder maps the stack of all propagated concepts to the output

$$P(Y \mid \mathbf{c}).$$

When the task output is itself a human-interpretable concept , this module may be omitted, while being necessary for tasks mapping back to a high-dimensional output space, e.g., sequence or video generation.

**Observation 4.1.** Note that the temporal resolution of concept propagation in the interpretable space does not necessarily need to match the input sampling rate. Depending on the model design choices, concepts may evolve at coarser or finer time scales, and mechanisms may advance time discretely (recurrent updates in the style of Hochreiter & Schmidhuber (1997)) or continuously (differential equations over concepts in the style of Chen et al. (2018)).

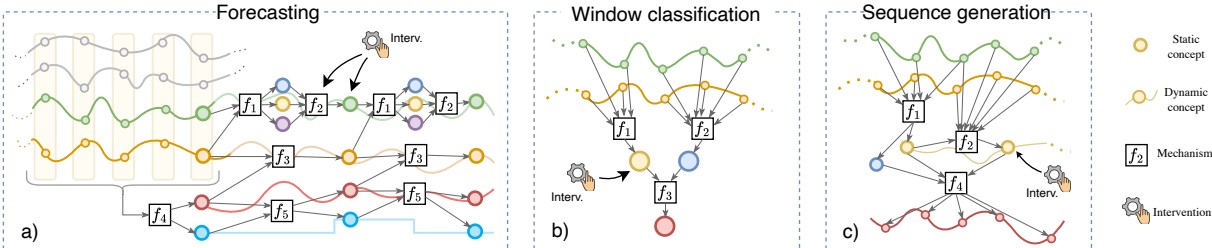

*Figure 3.* Examples of possible instantiations of the blueprint for different time series tasks. (a) Concept-based forecasting: some of the input variables (green and orange) are chosen as the forecasting target, while all available input information is used to encode new dynamic concepts that contribute to the forecasting. (b) Concept-based window classification: instantaneous concepts are extracted from different input subsequences and then combined to solve a downstream task. (c) Concept-based sequence generation: the first input subsequence is used to encode one instantaneous concept and one dynamic concept; later, the second subsequence is used to drive the evolution of the dynamic concept. Finally, all available interpretable information is used to generate a new time series.

### 4.3. How to enforce semantic alignment?

How to enforce SA in time series settings leaves the door open for many research directions. At present, the standard approach within the concept-based community consists of enforcing SA **during training** through **explicit supervision** of the concept variables with available ground truth labels. Building on this, we can speculate on a training objective combining three terms:

$$\mathcal{L} = \alpha \, \mathcal{L}_{\text{task}} + \beta \, \mathcal{L}_{\text{concept}} + \gamma \, \mathcal{L}_{\text{prop.}}$$

where:

- $\mathcal{L}_{\text{task}}(\hat{\mathbf{y}}, \mathbf{y})$ is the standard loss for task supervision;

- $\mathcal{L}_{\text{concept}} \left( \{\hat{\mathbf{c}}_t^{(k)}\}_{k \in \mathcal{S}}, \{\mathbf{c}_t^{(k)}\}_{k \in \mathcal{S}} \right)$ supervises the concept encoder by comparing predicted source concepts $\hat{\mathbf{c}}_t$ to their ground-truth labels $\mathbf{c}_t$. This term is responsible for the SA of the source concepts (Eq. 2);

- $\mathcal{L}_{\text{prop.}} \left( \{\hat{\mathbf{c}}_{t+1}^{(k)}\}_{k \in \mathcal{S} \cup \mathcal{D}}, \{\mathbf{c}_{t+1}^{(k)}\}_{k \in \mathcal{S} \cup \mathcal{D}} \right)$ supervises the propagation layers by comparing propagated concepts against ground-truth labels at future timesteps. This term enforces temporal consistency and preserves SA under concept evolution (Eq. 3),

and $\alpha, \beta, \gamma$ are the loss weighting hyperparameters. Omitting $\mathcal{L}_{\text{prop.}}$ generally allows concept drift over time, even when per-time-step concept predictions are accurate.

As discussed in Sec. 3, alignment at the level of mechanisms can be enforced by constraining the hypothesis space of admissible functions. Typically, this corresponds to imposing constraints derived from the user's prior knowledge, e.g., physical laws or mathematical properties. However, such approaches can severely reduce expressivity when applied to human-interpretable variables. We believe an important open direction is thus to explore how mechanism alignment could be achieved while preserving expressivity. One option

is to achieve it through regularization or auxiliary losses on otherwise universal approximators, yet this provides limited guarantees. We speculate that a more principled alternative is to design propagation mechanisms by composing primitive modules (i.e., layers) with known properties, yielding expressive modules that remain closed under composition and satisfy desired constraints by construction (You et al., 2017).

**Observation 4.2** (Compatibility with CBMs advances)**.** As this framework directly extends the CBM approach, it is trivially compatible with recent advances in the related literature, including probabilistic concepts (Vandenhirtz et al., 2024), concept spaces preserving expressivity of black-box models (Espinosa Zarlenga et al., 2022), finite-memory mechanisms (Debot et al., 2024), and loss regularizations.

## 5. Impact

### 5.1. Impact for time series: enabled properties

Models that satisfy the proposed blueprint enable a set of properties that we argue could be important for a wider adoption of deep time series models in high-stakes domains and scientific domains. First, semantically aligned models are **actionable**, defined as the possibility for humans to adjust both model variables and mechanisms in view of taking concrete actions (e.g., debugging, decision making). This happens naturally as model and user would operate at the same level of abstraction. SA further enables **verifiability**, as aligned representations and propagation mechanisms can be assessed against explicit, domain-relevant properties using formal verification techniques such as model checking (Baier & Katoen, 2008). This contrasts with relying solely on experimentally testable properties, such as predictive accuracy, which are notoriously insufficient to establish reliability in high-stakes applications (Sculley et al., 2015; Amodei et al., 2016; Rudin, 2019). Furthermore, when aligned concepts correspond to protected attributes or safety-critical states, their influence on predictions can be

explicitly traced over time, supporting **fairness** analysis in sequential decision-making settings (Goodman & Flaxman, 2017; Barocas et al., 2023). Moreover, constraining model behavior at the level of stable, domain-relevant concepts rather than raw observations promotes **robustness** under distribution shift, a central concern in time series applications (Ovadia et al., 2019; Benidis et al., 2022). Collectively, these properties establish conditions for **trustworthiness** here understood as warranted trust in the model's outcomes (Jacovi et al., 2021; Lee & See, 2004).

### 5.2. Impact for time series: new model interactions and design opportunities

**Test-time intervention and model adaptation.** Different types of interventions can be applied directly in the concept space and propagated forward through the learned temporal mechanisms. For example, correcting a concept value at time $t$ (e.g., revising a clinical state) results in an update of all future concept estimates without requiring repeated intervention at each step. This could also enable temporally consistent counterfactual analysis that is not available in models whose internal states are semantically opaque. We believe it is an open and promising direction to study how such targeted updates could be integrated with parameter updates in an online or continual learning setting, where models are incrementally adapted as new feedback becomes available (Hoi et al., 2021; Shi et al., 2025). Similarly, when a user identifies that a concept is systematically mispredicted by a learned mechanism, an architecture of this kind permits targeted interventions on the responsible mechanism, depending on the degree of mechanism alignment, rather than requiring retraining.

**Concept-conditioned generation and forecasting.** For generative time series tasks (e.g., sequence generation, reconstruction, data augmentation), aligned models allow generation to be conditioned on specified concept values or mechanism constraints (e.g., "generate a scenario where thermal stress increases linearly") (Yingzhen & Mandt, 2018; Ismail et al., 2024; 2025). In practice, we speculate this could open new directions for *virtual sensing* (Wu et al., 2021b; De Felice et al., 2024), where unobserved signals (e.g., due to cost, failures, or invasiveness) could be generated conditional on physically or semantically meaningful concepts, such as weather conditions, physiological states, or system health indicators. This perspective aligns with recent interest in conditional next-token prediction in LLMs, where text generation is guided by explicit intermediate representations or control variables (Li et al., 2023; Zhang et al., 2023; Sun et al., 2025).

**Interaction with human-designed knowledge.** SA enables interaction not only with humans, but also with human-

designed knowledge bases, and thus acts as a fundamental bridge between interpretable models and hybrid architectures, e.g., neurosymbolic approaches (Marra et al., 2024). This bridge can be leveraged to enrich interpretable models with alternative alignment mechanisms, such as formal constraints rather than supervision (e.g., $C_{\text{red}} \oplus C_{\text{green}}$), as well as symbolic components (e.g., symbolic propagation) and formal verification tools (e.g., concepts consistency checking via automated solvers). Recently, growing interest in neurosymbolic models that operate over time (Manginas et al., 2024; De Smet et al., 2025) has created a timely opportunity for cross-fertilization in the time series context.

### 5.3. Impact for concept-based models

This work highlights the largely unexplored problem of concepts whose semantics evolve over time. From a theoretical angle, it motivates extending recent formal frameworks for interpretability (Barbiero et al., 2025) to account for definitions and guarantees that hold across time. More broadly, the paper isolates mechanism alignment as a distinct and underexplored challenge in CBMs, as opposed to concept prediction accuracy, and draws parallels with constrained dynamical systems and physics-informed modeling as sources of methodological insight. Finally, viewing temporal evolution as a sequence of concept refinements suggests new directions for addressing concept leakage.

## 6. Alternative Views

We present here views that are opposed to our position, followed by our rebuttal to each. A more extensive literature review of existing interpretability paradigms, with each claim addressed independently, is provided in App. A, with a summary in Tab. 1.

**6.1 Semantic interpretability can be achieved post-hoc.** A widespread position in the literature states that flexible, post-hoc explanations are sufficient as long as they expose aspects of model internal reasoning, even when these are expressed at a different level of abstraction than that of the user. This view underlies a broad range of paradigms, including input attribution, counterfactuals, surrogates, LLM-based explanations, and mechanistic interpretability. **Rebuttal.** We argue that such approaches target a different form of opacity, i.e., structural opacity, rather than semantic opacity (Sec. 2.3). While these methods may reveal model internal computations or sensitivities, the burden of interpretation is shifted to the user, who must bridge the gap between the explanation and their own conceptual framework without any guarantees that such a bridge exists. In practice, these explanations may provide qualitative insights into model behavior, support hypothesis generation, and help identify spurious correlations or dataset biases, especially in ex-

ploratory settings. Yet, we argue this is insufficient for users in other domains to understand, validate, trust predictions, or to reliably intervene on the model. Importantly, explanation methods cannot provide formal guarantees on whether the extracted patterns are what is actually used by the deep model inference. The intuition is that the explanation model is a different model than the one used for predictions (note how this is not related to how interpretable the input space is). In support of this, there is established evidence both empirical (Adebayo et al., 2018) and theoretical (Bilodeau et al., 2024). The theoretical result holds irrespective of the input domain, and therefore also applies to time series data. A meeting point between explanations and interpretability-by-design is that explanations could, for example, be used for discovering patterns that might be relevant for concept discovery.

### 6.2 Mechanistic transparency is sufficient for semantic interpretability.

A second influential view equates interpretability with transparency of a model's mechanisms or dynamics. According to this position, models are considered interpretable if their computations are simple, constrained, or analyzable, e.g., linear latent dynamics, attention weights, symbolic equations, physically motivated constraints, or fixed equations. While these methods differ substantially in scope and assumptions, they all claim that constraining the hypothesis space or exposing internal structure suffices to render model reasoning interpretable. **Rebuttal.** While aligning mechanisms is necessary, mechanisms capture only part of a model's internal reasoning. For a model to be fully interpretable, SA must be ensured both at the level of mechanisms *and* variables (Sec. 3). Alignment of mechanisms alone is sufficient only when the input variables already correspond to the concepts relevant for reasoning. When inputs lack clear semantics, or when domain understanding relies on more abstract or derived concepts, as in physiological or financial time series, these approaches may fail to provide meaningful explanations.

### 6.3 Semantic alignment trades off against performance.

According to a common position, constraints introduced to make models interpretable necessarily reduce model expressivity. Under this view, highly accurate deep time series models are assumed to require high-dimensional representations, while interpretable models are seen as simplified approximations suitable only when performance requirements are relaxed. This position is often supported by empirical comparisons on standard benchmarks. Moreover, in many application domains, predictive accuracy is the primary criterion for deployment. **Rebuttal.** We believe that the accuracy-interpretability tradeoff is overstated. Prior work has established that concept-based models can match the performance of black-box architectures (Koh et al., 2020), and that concept embedding (Espinosa Zarlenga et al., 2022) and

*Table 1.* SA properties of existing interpretability paradigms. Inst.: SA of Instantaneous concepts, Dyn.: SA of Dynamic concepts, MA: Mechanism Alignment.

| Approach | Inst. | Dyn. | MA |
|---|---|---|---|
| *Post-hoc Methods* | | | |
| Input importance | ✗ | ✗ | ✗ |
| Surrogates | ✗ | ✗ | ✗ |
| Counterfactuals | ✗ | ✗ | ✗ |
| LLM explanations | ✗ | ✗ | ✗ |
| Mechanistic interpretability | ✗ | ✗ | ✗ |
| *Intrinsic Methods* | | | |
| Attention | ✗ | ✗ | ✗ |
| Koopman / Linearization | ✗ | ∼ | ∼ |
| Symbolic regression | ∼ | ∼ | ✓ |
| Prototype-based | ∼ | ✗ | ✗ |
| Physics-Informed | ∼ | ∼ | ✓ |
| Time series primitives | ∼ | ✗ | ∼ |

✓ = satisfies ∼ = partially satisfies ✗ = fails

residual (i.e., unconstrained) pathways can recover full expressivity while preserving interpretability (Mahinpei et al., 2021; Shang et al., 2024; De Felice et al., 2025). Moreover, in many applications, particularly in high-stakes settings, it remains unclear whether reported performance gains of deep learning reflect robust generalization or overfitting to benchmark datasets.

### 6.4 Concept annotations are too expensive.

A common objection to semantically aligned models is that obtaining concept annotations at scale is prohibitively expensive, particularly when supervision is required at every timestep for both spatial and temporal mechanisms. **Rebuttal.** While labeling from human experts remains the gold standard, recent trends in the CBM literature suggest that scalable alternatives are emerging. Automated approaches now span several complementary directions: LLMs can propose concepts and provide annotations (Oikarinen et al., 2023; Feng et al., 2026), concept discovery methods such as ACE (Ghorbani et al., 2019) can extract candidate concepts from pre-trained models, and mechanistic interpretability tools can identify and rank concepts directly from learned representations (De Santis et al., 2026). Furthermore, the use of formal constraints (Sec. 5.2) can reduce the burden of supervision by leveraging relational dependencies between concepts. The viability of these approaches at scale is increasingly supported by empirical evidence: industry efforts such as GuideLabs (GuideLabs, 2025) have developed scalable annotation workflows, and the recently released Steerling-8B model (GuideLabs, 2026) reports that a concept-based approach matches the downstream performance of LLMs trained on 2-7× more data. This suggests that semantic alignment and competitive accuracy are jointly attainable at scale. Beyond extending such static approaches, temporal dependencies in time series data can be exploited to

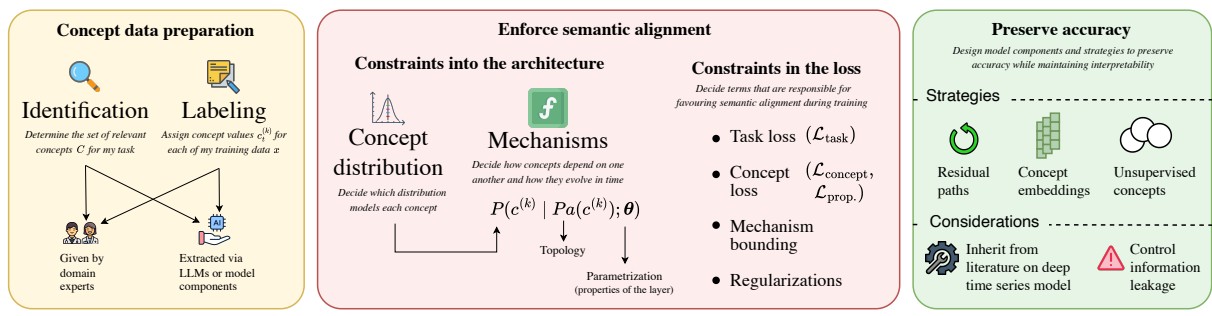

*Figure 4.* Practical guidelines to implement an interpretable deep time series model.

decrease the granularity of required labels, e.g., labeling every $n$-th timestep.

## 7. Practical Implementation Guidelines

In light of the proposed blueprint (Sec. 4) and the discussion of alternative views (Sec. 6), we now provide a compact recap of the concrete guidelines for implementing semantically aligned deep time series models, with a focus on practical challenges. As summarized in Fig. 4, we organize the guidelines along three axes: preparing the concept data, enforcing SA, and preserving predictive accuracy.

**Concept data preparation.** Before any modeling, the user should address:

- *Concept identification*: determine the set of concepts $\mathcal{C}$ relevant for the task.
- *Concept labeling*: assign values $c_t^{(k)}$ to each concept, associated with an appropriate subsequence of the observations $\mathbf{x}_{\leq t}$ at the temporal resolution at which the concept is defined (e.g., window or timestamp level).

For both, the more robust approach is to rely on domain experts. Alternatively, LLM-based annotation, concept discovery, as well as the other strategies discussed in Sec. 6.4, should be considered to make this tractable in practice. Additional challenges related to concept labeling include irregular, missing, and noisy values. These can be addressed by combining methods from multi-label time series classification (Guan et al., 2016; Zhang et al., 2022) (at least for discrete concepts) with recent developments in static interpretable architectures, e.g., probabilistic extensions (Vandenhirtz et al., 2024).

**Design of the interpretable space.** Control over the interpretable space is determined by two complementary classes of design choices:

- *Architectural constraints*: specify the concept distribution

family $P\big(c^{(k)} \mid \mathrm{Pa}(c^{(k)}); \boldsymbol{\theta}\big)$, the dependency topology among concepts, and the parametrization of encoding and propagation mechanisms (e.g., monotonic, linear, or rule-based layers).

- *Soft constraints in the loss*: include the three-term objective detailed in Sec. 4.3, optionally combined with regularizations to bound the mechanisms' hypothesis space to match user-defined qualitative constraints, e.g., monotonicity or sparsity.

The two classes are complementary: architectural constraints provide guarantees by construction, but could harm expressiveness, while soft constraints offer more flexibility at the cost of interpretability and generalization.

**Preserving accuracy.** When the specified concept set is incomplete (i.e., the specified concepts do not capture all task-relevant information), constraining predictions to pass through a concept bottleneck may degrade predictive performance. As anticipated in Sec. 6.3, three established strategies from the static CBM literature mitigate this: residual pathways (Yuksekgonul et al., 2023), concept embeddings (Espinosa Zarlenga et al., 2022), and unsupervised concepts (Shang et al., 2024). Since these operate along the non-interpretable path, advances in deep time series modeling can be inherited directly. Extending these strategies to the temporal setting, while controlling *information leakage*—the bypassing of concepts through residual paths, which is known to limit the effectiveness of interventions (Havasi et al., 2022)—is an open research direction.

## Concluding Remarks

We urge greater attention to the issue of semantic opacity. In proposing new models, including methods not targeting the interpretability community, we call for researchers not only to discuss accuracy metrics but to be clear about which types of alignments the model could guarantee, both at the level of variables and mechanisms, and how these alignments are ensured.

## Acknowledgements

This work is supported by the Swiss National Science Foundation (SNSF) through the grant 205121_197242 for the project "PROSELF: Semi-automated Self-Tracking Systems to Improve Personal Productivity", and by the Hasler Foundation through the project "Towards Scalable Multimodal Causal Deep Learning" (grant ID 2024-05-15-70). RD and AT acknowledge support from the Swiss State Secretariat for Education, Research and Innovation (SERI) under contract No. 24.00184 (AutoMoTIF project). The project was selected under the EU Horizon Europe programme, grant agreement No. 101147693. PB acknowledges support from the SNSF, through the project "IMAGINE" (grant ID 224226). GM and PB acknowledge support from the Research Foundation Flanders, through the project "Relational Concept-Based Models" (grant ID G033625N). GM also acknowledges the Flemish Government under the "Onderzoeksprogramma Artificiële Intelligentie (AI) Vlaanderen" programme.

The views and opinions expressed are, however, those of the authors only and do not necessarily reflect those of the funding agencies, which cannot be held responsible for them. We thank the anonymous reviewers for their valuable feedback, which substantially improved this work.

## Impact Statement

This paper presents work whose goal is to advance the field of Machine Learning. There are many potential societal consequences of our work, none which we feel must be specifically highlighted here.

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

# A. Current paradigms (extended)

A wide range of methods have been proposed to address the opacity of deep time series models (Rojat et al., 2021; Zhao et al., 2023). Although often grouped under the labels of "interpretability" or "explainability", these two terms should not be used interchangeably as they differ substantially in the guarantees they offer to end users. In this section, we critically examine existing paradigms by identifying the explainability/interpretability claims they advance, evaluating their assumptions, and the abstraction level at which these claims are realized, with a focus on whether they satisfy semantic alignment. We organize current paradigms into two categories: *post-hoc explainability methods* and *intrinsic (ante-hoc) interpretability methods*.

## A.1. Post-hoc explanations

**Input importance as explanation.** A large body of work frames interpretability as identifying which input variables or timesteps most influence a model's prediction. Feature attribution-, saliency- and masking-based methods all instantiate this idea by assigning importance scores to individual features, timesteps, or temporal windows (Zhao et al., 2023). Several adaptations have been proposed specifically for time series models, including Temporal Saliency Rescaling (Ismail et al., 2020), FIT (Tonekaboni et al., 2020), TimeSHAP (Bento et al., 2021), WinIT (Rooke et al., 2021), Dynamask (Crabbé & Van Der Schaar, 2021), and WindowSHAP (Nayebi et al., 2023). While such methods can be useful for debugging or exploratory analysis, input importance reduces the whole reasoning to a scalar notion (e.g., "this time window was important"), rather than to domain-relevant concepts, and is therefore often difficult for humans to interpret, compare, or translate into concrete actions (e.g., humans would not know what to change to obtain a different prediction).

**Model extraction and surrogate explanations.** Another post-hoc paradigm claims interpretability by approximating a trained model with a simpler surrogate. In time series settings, this includes extracting symbolic rules or motifs that mimic the original model's behavior. Methods such as MEME (Kazhdan et al., 2020), LASTS (Guidotti et al., 2020), TS-MULE (Schlegel et al., 2021), LIMESegment (Sivill & Flach, 2022), and TimeX/TimeX++ (Queen et al., 2023; Liu et al., 2024) fall into this category. From a semantic alignment perspective, this introduces an additional failure mode compared to feature attribution methods. Not only are explanations expressed in terms of input-level features or latent representations rather than domain-level concepts, but they are also mediated by an approximate model that effectively adds an additional layer of opacity. As a result, neither semantic alignment of variables nor alignment of mechanisms is guaranteed.

**Counterfactual explanations.** Counterfactual explanations aim to identify minimal changes to an input that would alter a model's prediction. A few methods extend this paradigm to time series by constructing alternative trajectories that flip the predicted label (Ates et al., 2021; Delaney et al., 2021). Compared to saliency methods, counterfactuals are often presented as more actionable, as they explicitly specify changes that would cross the model's decision boundary. However, existing time-series counterfactuals (including the ante-hoc CounTS (Yan & Wang, 2023) model) are expressed in terms of numerical perturbations of input features, which, in general, do not necessarily correspond to the abstract concepts through which users reason.

**LLM-based natural language explanations.** Recent approaches use LLMs to generate textual explanations of time series predictions (Jin et al., 2024b; Zhao et al., 2025; Cao et al., 2025). While these explanations use domain-specific vocabulary, they are typically generated post hoc and are not grounded in the internal variables and mechanisms of the predictive model. As a result, they may appear semantically rich while remaining unfaithful to the model's actual reasoning process.

**Mechanistic interpretability.** A recent line of work applies mechanistic interpretability tools to time series models, particularly time series foundation models. One direction analyzes internal representations, circuits, or subspaces to identify components that appear to correspond to interpretable patterns or concepts (Kalnāre et al., 2025; Pandey et al., 2025; Bao et al., 2026; Zou et al., 2025). A complementary direction leverages sparse autoencoders to decompose learned representations into sparsely activating, candidate-interpretable features (Wiliński et al., 2025; Divo et al., 2026; Oublal et al., 2026). While valuable for inspecting model internals, these methods operate post hoc and provide no structural guarantees that the identified components align with domain-relevant concepts: any observed correspondence is incidental to the training objective rather than enforced by the architecture.

## A.2. Intrinsic interpretability

**Attention mechanisms.** Attention mechanisms are frequently presented as interpretable under the assumption that attention weights can reveal a model's focus on the input timesteps and features (Choi et al., 2016; Lim et al., 2021; Zhou et al., 2021). However, as extensively discussed in the literature (Jain & Wallace, 2019; Wiegreffe & Pinter, 2019; Liu et al., 2022), attention weights do not, in general, provide reliable explanations of model reasoning. In time series models, attention operates over learned embedding spaces and time indices, and semantic alignment with domain-relevant concepts is not explicitly enforced; as a result, attention weights may fail to correspond to the concepts experts use for reasoning.

**Linearization.** Another class of approaches seeks interpretability by simplifying the learned dynamics. For example, Koopman-based models (Lusch et al., 2018; Brunton & Kutz, 2022; Kumar et al., 2024; Guerra et al., 2024) learn a latent space where the dynamics is linear. However, this requires a mapping to high-dimensional representations that lack semantics in the general case.

**Prototype-based networks.** Prototype-based models explain predictions by reference to representative examples learned during training (Gee et al., 2019; Ghods & Cook, 2022; Obermair et al., 2023; Huang et al., 2025). While this enables a form of case-based reasoning, the association typically relies on a (unsupervised) similarity in a learned embedding space. As a result, the model does not manifest *why* a given prototype is considered similar or relevant, nor which properties of the input justify the association. The burden of interpreting such associations is thus shifted to the user.

**Physics-informed and symbolic models.** Physics-Informed Neural Networks incorporate physical constraints during model training, either as fixed equations or more general constraints such as properties or symmetries (Raissi et al., 2019; Sholokhov et al., 2023; Horn et al., 2025). This approach can provide strong interpretability guarantees when the relevant physical laws are well specified. Similarly, symbolic regression approaches, including SINDy (Brunton et al., 2016) and Kolmogorov–Arnold Networks (Xu et al., 2024; Barašin et al., 2025; Somvanshi et al., 2025), aim to learn closed-form mechanisms governing the evolution of the system. Despite their differences, both paradigms rely on the same strong assumption: the input variables already correspond to the concepts relevant for reasoning. As a result, they primarily address alignment at the level of mechanisms, while taking semantic alignment of variables for granted. When the inputs lack semantics or when the domain understanding depends on more abstract or derived concepts (e.g., physiological signals or financial time series), these approaches do not satisfy semantic alignment.

**Time series primitives.** Some intrinsic approaches, such as N-BEATS (Oreshkin et al., 2020), ShapeNet (Li et al., 2021), BasisFormer (Ni et al., 2023), and TIMEVIEW (Kacprzyk et al., 2024), seek interpretability by decomposing predictions into structured primitives commonly used in time series analysis (e.g., trend, seasonality, shapelets, motifs, splines, extrema) and explicitly composing the prediction from them [4]. These approaches come closer to our desiderata by constraining predictions to be mediated by a set of explicit intermediate concepts, rather than by opaque latent states. However, the resulting decompositions rely on a fixed, largely domain-agnostic set of statistical components that does not necessarily correspond to the concepts through which domain experts reason.

# B. Concrete instantiation

## B.1. Template architecture

We provide an example on how a Transformer Encoder would fit into the definitions of Sec. 2.2.

- The token embedding + positional encoding acts as Enc() (mapping each observation to a representation).

- The stack of self-attention + feed-forward layers is the Prop() (each layer refines representations by exchanging information across time, eventually with causal masking).

- Eventually, a separate task head serves as Dec().

---

[4] Here, we do not consider opaque methods that use such primitives only as an internal inductive bias, e.g., Wu et al. (2021a); Zhou et al. (2022)

### B.2. Concept-based Deep Time Series model for system monitoring

We provide a concrete instantiation below for the running example we use in the paper, alongside the corresponding non-interpretable architecture.

An engineer monitoring an industrial system reasons about failure through three concepts: *overheating* $c_t^{\text{heat}} \in \{0,1\}$ (temperature exceeding a critical threshold), *thermal stress* $c_t^{\text{stress}} \in \mathbb{R}_{\geq 0}$ (cumulative heat exposure), and *degradation* $c_t^{\text{deg}} \in \mathbb{R}_{\geq 0}$ (stress-induced component wear).

|  | **Deep TS (SSM)** | **Interpretable** |
|---|---|---|
| **Data** | $\{x_t, y_t\}$ | $\{x_t, y_t, c_t\}$ |
| **Enc** | $h_t = \text{TCN}(x_{\leq t})$ | $\hat{c}_t^{\text{heat}} = \mathbf{1}[x_t > \alpha]$ |
|  |  | $\hat{c}_t^{\text{stress}} = \text{TCN}(x_{\leq t})$ |
| **Prop** | $h_{t+1} = \text{MLP}(h_t)$ | $\hat{c}_{t+1}^{\text{stress}} = g_\phi(\hat{c}_t^{\text{stress}})$ |
|  |  | $\hat{c}_{t+1}^{\text{deg}} = f_\theta(\hat{c}_t^{\text{stress}})$ |
| **Dec** | $\hat{y}_{t+1} = \text{Lin}(h_{t+1})$ | $\hat{y}_{t+1} = \text{Lin}(\hat{c}_t^{\text{heat}}, \hat{c}_{t+1}^{\text{stress}}, \hat{c}_{t+1}^{\text{deg}})$ |
| **Losses** | $\mathcal{L}_{\text{task}}(\hat{y}_{t+1}, y_{t+1})$ | $\mathcal{L}_{\text{task}}(\hat{y}_{t+1}, y_{t+1})$ |
|  |  | $\mathcal{L}_{\text{conc}}(\hat{c}_t^{\text{heat}}, c_t^{\text{heat}}, \hat{c}_t^{\text{stress}}, c_t^{\text{stress}})$ |
|  |  | $\mathcal{L}_{\text{prop}}(\hat{c}_{t+1}^{\text{stress}}, c_{t+1}^{\text{stress}}, \hat{c}_{t+1}^{\text{deg}}, c_{t+1}^{\text{deg}})$ |

A concept encoder maps raw temperature readings to source concepts, and a propagation module evolves them forward in time. Crucially, mechanism alignment (Def. 2) could be enforced architecturally: $g_\phi$ could be constrained to be monotonically non-decreasing, to reflect the irreversibility of cumulative thermal damage, and $f_\theta$ could be made monotone in 'stress'. Such constraints can be realized, e.g., via non-negative parametrizations. A linear decoder then maps the concept state to 'failure' logits.

As mentioned above, residual paths (Obs. 4.3) can be implemented to restore black-box performance if needed.

