# OpenReview forum: "Position: Interpretability in Deep Time Series Models Demands Semantic Alignment"
_ICML.cc/2026/Position_Paper_Track — ICML 2026 Position Paper Track regular_

### Official Review · Reviewer_wZGK · 2026-03-10

**Significance:** 3
**Argument Clarity:** 3
**Rating:** 5
**Confidence:** 4

**Questions:**

1. Some statements are not clearly presented. For example, the formula in L244 seems also implying some independence assumption. Is it reasonable, and why?
2. It is not clear how to fullfil the semantic alignment via the combination of the three loss terms in Section 4.3 in practice. Does it require extra annotation on the time series samples? What about the potential implementation costs and challenges?
3. Is it possible to give a running example which follows the suggested blueprint? It would be more persuasive if an examplar time series model is shown.

**Alternative Views Section:**

Yes

**Compliance With Llm Reviewing Policy A Conservative:**

Affirmed.

**Discussion Potential:**

3

**Final Justification:**

Since that the responses in the rebuttal have fully solved my concerns, I would like to update my rating to an accept.

**Paper Summary:**

The paper states a position to urge greater attention to semantic opacity in deep time series models and call for the community not only discuss the accuracy metrics but pay attention to semantic alignment at both the levels of variables and mechanisms and how the alignment being ensured. Moreover, the paper presented a set of requirements for the model design and the model properties to develop semantically aligned deep time series models.

**Position:**

Yes

**Position In Title:**

Yes

**Related Work:**

4

**Strengths And Weaknesses:**

Strengths:
+ It is of important value to enforce semantic alignment (SA) in deep time series models, which enables the models to use variables and mechanisms corresponding to those used by domain experts.
+ The literature review is thorough and insightful, and the opacity problem is analysized clearly.
+ The semantic alignment is divided into semantic alignment of concept and semantic alignment of mechanism, both of them are clearly formalized.
+ It sounds making a lot of sense by introducing a structured concept space, in which concepts are propagated by aligned mechanisms both across space and time.

Weaknesses:

- Some statements are not clearly presented. For example, the formula in L244 seems also implying some independence assumption. Is it reasonable, and why?
- It is not clear how to fullfil the semantic alignment via the combination of the three loss terms in Section 4.3 in practice. Does it require extra annotation on the time series samples? What about the potential implementation costs and challenges?
- Is it possible to give a running example which follows the suggested blueprint? It would be more persuasive if an examplar time series model is shown.

**Support:**

3

---

> ### Author Rebuttal · Authors · 2026-03-30
>
> We thank the reviewer for their positive feedback. We address the main points below.
>
> ---
>
> **W1/Q1 - Conditional independence**
>
> > the formula in L244 seems also implying some independence assumption. Is it reasonable, and why?
>
> The factorization $P(Y, \mathbf{c} \mid \mathbf{x}) = P(Y \mid \mathbf{c}) \cdot P(\mathbf{c} \mid \mathbf{x})$ follows from the chain rule under a **conditional independence assumption**: $Y \perp X \mid \mathbf{c}$, i.e., all task-relevant information flows through the concept bottleneck. This is a **design choice**, adopted, for example, in Concept Bottleneck Models [1]. We will clarify this in the revised text to avoid ambiguity.
>
> Note that this assumption holds only when the set of concepts is *complete* w.r.t. task, meaning it contains all the information to solve it. In all other cases, this would restrict the available information and the interpretability provided by concepts would inevitably trade off accuracy. Current research provides strategies to relax this assumption in static settings via concept embeddings, residual paths, and unsupervised concepts (also see response **W1/Q1 to Reviewer 1j78**). Relaxing this assumption in dynamic settings is what we advocate for interesting future research directions.
>
> ---
>
> **W2/Q2 - Clarification on SA**
>
> > Does it require extra annotation on the time series samples? What about the potential implementation costs and challenges?
>
> The reviewer is correct: SA generally requires ground-truth concept labels (**Sec. 6.4**). These supervise the model’s internal concept variables. The three loss terms then serve to supervise each of the three different modules we discuss (*Enc()*, *Prop()*, *Dec()*) individually.
>
> Such reliance on concept annotations could lead to practical challenges in real-world scenarios (**Sec. 6.4**). We kindly point the reviewer to our response **W1/Q1 to Reviewer 1j78** for details.
>
> These overheads are justified in favor of interpretability, robustness [2], and sample efficiency. On the latter, about a month ago, the Steerling-8B model (https://www.guidelabs.ai/post/steerling-8b-base-model-release/) showed that its concept-based approach could achieve downstream performance within the range of LLMs trained on $2-7 \times$ more data.
>
> ---
>
> **W3/Q3 - Practical example**
>
> > Is it possible to give a running example which follows the suggested blueprint?
>
> We provide a concrete instantiation below for the running example we use in the paper, alongside the corresponding non-interpretable architecture.
>
> An engineer monitoring an industrial system reasons about failure through three concepts: *overheating* $c^{\text{heat}}\_t \in$ {0,1} (temperature exceeding a critical threshold), *thermal stress* $c^{\text{stress}}\_t \in \mathbb{R}\_{\geq 0}$ (cumulative heat exposure), and *degradation* $c^{\text{deg}}\_t \in \mathbb{R}\_{\geq 0}$ (stress-induced component wear).
>
> |  | **Deep TS (SSM)** | **Interpretable** |
> |:---:|:---:|:---:|
> | **Data** | {$x\_t, y\_t$} | {$x\_t, y\_t, c\_t$} |
> | **Enc** | $h\_t = \mathrm{TCN}(x\_{\leq t})$ | $\hat{c}\_t^{\text{heat}} = \mathbf{1}[x\_t > \alpha]$ |
> | | | $\hat{c}\_t^{\text{stress}} = \mathrm{TCN}(x\_{\leq t})$ |
> | **Prop** | $h\_{t+1} = \mathrm{MLP}(h\_t)$ | $\hat{c}\_{t+1}^{\text{stress}} = g\_\phi(\hat{c}\_t^{\text{stress}})$ |
> | | | $\hat{c}\_{t+1}^{\text{deg}} = f\_\theta(\hat{c}\_t^{\text{stress}})$ |
> | **Dec** | $\hat{y}\_{t+1} = \mathrm{Lin}(h\_{t+1})$ | $\hat{y}\_{t+1} = \mathrm{Lin}(\hat{c}\_t^{\text{heat}}, \hat{c}\_{t+1}^{\text{stress}}, \hat{c}\_{t+1}^{\text{deg}})$ |
> | **Losses** | $\mathcal{L}\_{\mathrm{task}}(\hat{y}\_{t+1}, y\_{t+1})$ | $\mathcal{L}\_{\mathrm{task}}(\hat{y}\_{t+1}, y\_{t+1})$ |
> |  |  | $\mathcal{L}\_{\mathrm{conc}}(\hat{c}^{\text{heat}}\_t, c^{\text{heat}}\_t, \hat{c}^{\text{stress}}\_t, c^{\text{stress}}\_t)$ |
> |  |  | $\mathcal{L}\_{\mathrm{prop}}(\hat{c}^{\text{stress}}\_{t+1}, c^{\text{stress}}\_{t+1}, \hat{c}^{\text{deg}}\_{t+1}, c^{\text{deg}}\_{t+1})$ |
>
> A concept encoder maps raw temperature readings to source concepts, and a propagation module evolves them forward in time. Crucially, mechanism alignment (Def. 2) could be enforced architecturally: $g_\phi$ could be constrained to be monotonically non-decreasing, to reflect the irreversibility of cumulative thermal damage, and $f_\theta$ could be made monotone in 'stress'. Such constraints can be realized, e.g., via non-negative parametrizations. A linear decoder then maps the concept state to 'failure' logits.
>
> As mentioned above, residual paths (Obs. 4.3) can be implemented to restore black-box performance if needed.
>
> We thank the reviewer for this, we will add the example to the paper.
>
> ---
>
> We hope to have clarified all of the reviewer’s concerns and are happy to provide further details if needed.
>
> ---
>
> **Refs:**
> 1. Koh et al., Concept bottleneck models, ICML, 2020.
> 2. Barbiero et al., Relational concept bottleneck models, NeurIPS, 2024.

---

> > ### Author Rebuttal · Reviewer_wZGK · 2026-04-02
> >
> > The responses in the rebuttal have fully solved my concerns. Thus, I would like to keep the positive rating.

---

### Official Review · Reviewer_8NQh · 2026-03-14

**Significance:** 3
**Argument Clarity:** 2
**Rating:** 5
**Confidence:** 3

**Questions:**

- Q1: Regarding ll. 212f: Would it also be possible that the nonlinear "Prop" function entirely drops it in one propagation step, instead of "only" decaying exponentially over time?
- Q2: How would an architecture such as a Transformer Encoder fit into the definitions of Sec. 2.2? Generally, a section instantiating this definition for typical architectures (in the Appendix) might be useful.

**Alternative Views Section:**

Yes

**Compliance With Llm Reviewing Policy A Conservative:**

Affirmed.

**Discussion Potential:**

3

**Final Justification:**

The responses and other reviews have not raised any new concerns. Most concerns were addressed to the degree necessary. Thus, I keep the positive rating.

**Paper Summary:**

The work first observes that while interpretability is highly desirable in many time series applications, it is hindered by the structural and semantic opacity of typical deep learning models. It argues that the semantic alignment of the model's internal concepts and mechanisms with those used by experts in the specific domain is crucial for proper interpretability. Derived from formal definitions, a framework for building interpretable deep time series models is proposed. It is argued that any method that does not explicitly enforce semantic alignment, i.e., the majority of existing work, will not ultimately achieve interpretability sufficient for high-stakes applications.

**Position:**

Yes

**Position In Title:**

Yes

**Related Work:**

2

**Strengths And Weaknesses:**

**Strengths:**
- The issue discussed is important, as there is indeed a lack of wholistically interpretable models for many time series tasks.
- The writing is mostly clear and of high quality.
- The formal definition is mostly sensible, and helps structure the arguments and future discussion/implementation of the proposed framework. It nicely covers many time series tasks, not just classification or forecasting.
- The discussion of alternative views is convincing and covers major counterpositions.

**Weaknesses:**
- The framework itself is not groundbreaking, while the focus is more on discussing why such a model should be used. Therefore, this is still suitable as a position paper.
- The discussion of how exactly the different existing methods fall short of proper interpretability (cf. Tab. 1) is deferred to the appendix. This is key to the work and should be given greater prominence.
- It is somewhat unclear what concretely a mechanism would be (Sec. 3.1 and 3.2)
- Regarding Def. 2: It is somewhat unclear how the "set of distributions admissible under human constraints" would be constructed in practice, given that its probabilistic nature might be challenging for human reasoning. Sec. 4 does not pick this up again.
- Fig. 3 needs improvement: The interventions mentioned are not discussed elsewhere. Furthermore, it is not sufficiently clear what the white boxes $f_1$, $f_2$, ..., etc. are. A legend might help. Which concepts are instantaneous and which are dynamic? Where are the mechanisms?

**Minor opportunities for improvement:**
- Fig. 1: Making explicit that the propagation step goes across time would make things easier to grasp at a glance.
- While Observation 2.2 is correct, I did not understand what it contributes to the exposition.
- Regarding Sec. 3.1: The definition of Mechanisms is disconnected (from the Concepts paragraph) at this point in the paper.
- Regarding Sec. 3.2, Instantaneous concepts: The key property here seems to be that they are derived from $\mathbf{u}$ instead of $\mathbf{z}$, i.e., not that their *future* values are irrelevant but rather that the *past* values are not necessary for identifying it.
- In Sec. 4.3, $\mathcal{L}$ is first defined via $\mathcal{L}\_\text{spatial}$ and $\mathcal{L}\_\text{temporal}$, but directly after that, they get renamed to $\mathcal{L}\_\text{comcept}$ and $\mathcal{L}\_\text{prop.}$.
- L. 331 contains a broken reference.
- Fig. 1 and 3 are never referenced.
- Fig. 3: Typo in caption: "Es." -> "Ex." (3x).
- Tab. 1: Abbreviating "SA of Instantaneous concepts" with "Inst." instead of "Ist." might be easier. The $\sim$ in the row "Prototype-based" is red instead of orange.

**Support:**

3

---

> ### Author Rebuttal · Authors · 2026-03-30
>
> We thank the reviewer for their positive feedback and suggestions. We address the main points below.
>
> ---
>
> **W2 - Literature**
>
> > discussion of [...] existing methods [...] is deferred to the appendix. [...] should be given greater prominence.
>
> **We agree** with this intent. Due to **space constraints**, we grouped methods by their primary limitation in the main text, and deferred the extensive presentation to the appendix.
>
> In case of acceptance of the paper, we will be able to utilize the additional page to revise this choice.
>
> ---
>
> **W3 - Clarification on Mechanisms**
>
> > It is somewhat unclear what concretely a mechanism would be
>
> We agree that the discussion of mechanisms can be deepened. In the revision, we will integrate the following clarification in Sec. 3.1.
>
> By *mechanisms*, we refer to **functional relations between concepts**, modeled as conditional distributions (CPD) (**Sec. 3.1**):
> $$P(\cdot \mid \cdots),$$
> i.e., how the probability of a concept changes given evidence on its parent concepts in space and time.
>
> A slight change in notation makes it more evident how these can be operationalized: we explicit the **parametric dependence** by
> $$P(\cdot \mid \cdots ; \boldsymbol{\theta}).$$
>
> There is flexibility in how the parametrization is realized in practice. A standard approach is to parameterize these CPDs with neural network weights, which would model the distribution (hence the dependences) implicitly. An arguably more interpretable setup is to assume that the family of a CPD is known and use a neural network to predict its parameters given the parent concepts.
>
> A practical example would be a layer predicting the parameters of a categorical distribution (i.e., concept activation probabilities) or, for continuous concepts, a layer predicting the mean and variance of a Normal distribution ($N(\mu=NN_1(x), \sigma=NN_2(x))$).
>
> Note how deterministic mechanisms are recovered as a particular case with delta distributions ($\delta(NN(x))=NN(x)$, footnote 2 at Page 3).
>
> We thank the reviewer for raising this point, which will strengthen the paper.
>
> ---
>
> **W4 - Hypothesis functional space**
>
> > It is somewhat unclear how the "set of distributions admissible under human constraints" would be constructed in practice
>
> In practice, humans can specify $\mathcal{M}$ directly with the family of admissible CPDs for a concept (see response above).
>
> Complementary, humans can specify **constraints on functional properties**. Such constraints act on the parametrization of the conditional distributions, which in turn induce $\mathcal{M}$. For example, one may restrict it to a particular class of layers (e.g., monotonic or linear). Different users will therefore induce different $\mathcal{M}$, depending on their domain knowledge.
>
> Note that how to guarantee general functional properties while maintaining the flexibility and expressiveness of deep learning is still an open research question (**Sec. 4.3 ll. 311-327**).
>
> ---
>
> **W5 - Legend for Fig. 3**
>
> > A legend might help
>
> Thank you for pointing this out. We will add a legend ('static concepts', 'dynamical concepts', mechanisms $f$, and interventions). Please find it at https://anonymous.4open.science/r/rebuttal_figures-D5D4/figures.png.
>
> ---
>
> **Minor point 4 - Clarification on instantaneous concepts**
>
> > The key property here seems to be [...] not that their future values are irrelevant but rather that the past values are not necessary for identifying it.
>
> We would like to clarify that the distinction does not lie in input dependency (both concept types may depend on $x_{\leq t}$). Rather, the difference lies in whether their temporal evolution (at $t+1$) is well-defined and inferable from observations up to time $t$, and whether modeling this evolution is relevant for the task. *Instantaneous* concepts represent properties of the system up to time $t$, i.e., they act as “snapshots”.
>
> ---
>
> **Q1 - Quick SA drop over time**
>
> > Would it also be possible that the nonlinear "Prop" function entirely drops it in one propagation step [...]?
>
> Yes, if semantic alignment is not explicitly enforced to be preserved over time, it is possible to lose it in one step (depending on overfitting of instantaneous alignment). That corresponds to a decay timescale shorter than one discrete timestep.
>
> ---
>
> **Q2 - Practical instantiation of a Transformer Encoder**
>
> > How would an architecture such as a Transformer Encoder fit into the definitions of Sec. 2.2?
>
> This is indeed possible. An example looks as follows:
>
> - The token embedding + positional encoding acts as *Enc()* (mapping each observation $x_t$ to a representation $u_t$).
> - The stack of self-attention + feed-forward layers is the *Prop()* (each layer refines representations by exchanging information across time, eventually with causal masking).
> - Eventually, a separate task head serves as *Dec()*.
>
> ---
>
> We hope to have clarified all of the reviewer’s concerns and are happy to provide further details if needed.

---

> > ### Author Rebuttal · Reviewer_8NQh · 2026-04-04
> >
> > I thank the authors for their response. The referenced figure and legend are a useful addition.
> >
> > Notes:
> > - Re. Q2: With Transformer *Encoder*, I meant one w/o causal masking, i.e., with full forward and backward attention. To my understanding, this does *not* fit into the described formalism.

---

### Official Review · Reviewer_1j78 · 2026-03-21

**Significance:** 3
**Argument Clarity:** 3
**Rating:** 4
**Confidence:** 3

**Questions:**

My biggest concern is practical usage. The authors point out a route to try, but many practical issues in real-world data are not discussed.

**Alternative Views Section:**

Yes

**Compliance With Llm Reviewing Policy A Conservative:**

Affirmed.

**Discussion Potential:**

2

**Final Justification:**

Practical implementation is further discussed.

**Paper Summary:**

This paper points out a desired property of interpretability in deep time series models: semantic alignment (SA). SA enables variables and internal model inference states (e.g., hidden variable states) to correspond to domain experts' reasoning concepts. A motivating example is that "clinicians may struggle to reason in terms of 'timestep 47', but can relate to constructs such as 'onset of tachycardia'."

This paper depicts the methodology of SA under an encoding–propagation–decoding time series model template, which is considered generic. The interpretability problem and corresponding concepts are systematically examined. SA of "concepts" and SA of "mechanism" are formally defined.

**Position:**

Yes

**Position In Title:**

Yes

**Related Work:**

3

**Strengths And Weaknesses:**

The paper is well-structured. Readers can follow the concepts fluently. The blueprint is presented in a clear mathematical way. However, the practical implementation is not very clear. Real data often lacks ground-truth concept labels and contains many missing entries. How should we prepare the data? What are the potential challenges? These questions are only partially addressed in the alternative views section.

Typo: Please check the notation in the first equation in Section 4.3.

**Support:**

3

---

> ### Author Rebuttal · Authors · 2026-03-30
>
> We thank the reviewer for their feedback and for pointing out the typo. We address the main points below.
>
> ---
>
> **W1 / Q1 - Practical usage**
>
> > My biggest concern is practical usage. [...] many practical issues in real-world data are not discussed.
>
> We acknowledge that the paper would benefit from a more systematic presentation of implementation guidelines. In response, we will **extend Sec. 4 with an additional subsection (Sec. 4.4, Practical Guidelines)**, integrating the three key aspects outlined below. This need was raised across reviewers, and we hope to address it with this addition.
>
> A compact graphical overview (https://anonymous.4open.science/r/rebuttal_figures-D5D4/figures.png) will be included.
>
> **Concept data preparation.**
> Before any modeling, the practitioner must:
> - identify the set of concepts $ \mathcal{C} $ relevant for the task,
> - assign values $ c^{(k)}\_t $ to these concepts, each associated with an appropriate subsequence of the observations $\mathbf{x}\_{\leq t}$, depending on the temporal resolution at which each concept is defined (e.g., window or timestamp level).
>
> For both, the more robust approach is to rely on domain experts. However, as one could imagine, this may be demanding in practice. To reduce this burden, LLMs can propose concepts and provide automated labeling (e.g., [1,2]), and concept discovery from pretrained models (e.g., ACE [3], or mechanistic interpretability) can extract and rank concepts from learned representations.
>
> While these considerations extend naturally to the temporal setting, an additional difficulty is that annotations are required across multiple timestamps. Concurrently, the temporal structure offers ways to alleviate this: temporal dependencies in $ \mathbf{x}_{0:T} $ can be exploited to reduce granularity of the annotations (e.g., labeling every $ n $-th timestep), and annotations can be obtained from auxiliary data sources that operate at the same level of abstraction as the concepts (e.g., sensor measurements of physical variables).
>
> Additional practical challenges (e.g., irregular / missing data, or partial / noisy labels) can be faced by combining methods from the multi-label time series classification settings [4,5] (at least for discrete concepts), with recent developments in static interpretable architectures, e.g., probabilistic extensions [6].
>
>
> **Enforcing SA.** Control over the interpretable space is determined by two classes of design choices:
>
> - Architectural constraints could specify the concept distribution family $ P\big(c^{(k)} \mid \mathrm{Pa}(c^{(k)}); \theta\big) $, the dependency topology among concepts, and the parametrization of encoding and propagation mechanisms (see response **W4 to Reviewer 8NQh**).
>
> - Soft constraints in the loss include the three-term objective detailed in **Sec. 4.3**, optionally combined with regularizations to bound the mechanisms' hypothesis space to match user-defined qualitative constraints (e.g., monotonicity, sparsity).
>
>
> **Preserving accuracy.** A key practical concern is that constraining predictions to pass through a concept bottleneck may degrade performance when the concept set is incomplete. Three established strategies could mitigate this:
>
> - residual pathways [7],
> - concept embeddings [8],
> - unsupervised concepts that let the model discover additional (latent) variables [9].
>
> These are well-studied in static CBMs; extending them to the temporal setting is an open direction we flag explicitly. Since this concerns the non-interpretable path, advances in deep time series models could be inherited directly.
>
> However, this introduces *information leakage*, where input information bypasses concepts through such residual paths. This is known to limit the effectiveness of interventions, leaving interesting research directions.
>
> We thank the reviewer for this point, which will strengthen the paper.
>
> ---
>
> We hope to have clarified all of the reviewer’s concerns and are happy to provide further details if needed.
>
> ---
>
> **Refs:**
> 1. Oikarinen et al., Label-free concept bottleneck models. ICLR, 2023.
> 2. Feng et al., Bayesian concept bottleneck models with llm priors. NeurIPS, 2025.
> 3. Ghorbani et al., Towards automatic concept-based explanations. NeurIPS, 2019.
> 4. Guan et al., Efficient multi-instance learning for activity recognition from time series data using an auto-regressive hidden markov model, ICML, 2016.
> 5. Zhang et al., Graph-guided network for irregularly sampled multivariate time series, ICLR, 2022.
> 6. Vandenhirtz et al., Stochastic concept bottleneck models. NeurIPS, 2024.
> 7. Yuksekgonul et al., Post-hoc concept bottleneck models, ICLR, 2023.
> 8. Espinosa Z. et al., Concept embedding models: Beyond the accuracy-explainability trade-off, NeurIPS, 2022.
> 9. Shang et al., Incremental residual concept bottleneck models, CVPR, 2024.
> ---

---

> > ### Author Rebuttal · Reviewer_1j78 · 2026-04-03
> >
> > Thanks for your detailed response. I‘m happy to increase my score.

---

### Official Review · Reviewer_hdEe · 2026-03-23

**Significance:** 3
**Argument Clarity:** 2
**Rating:** 4
**Confidence:** 4

**Questions:**

In principle, and speculatively, can the authors argue that a deep learning model can be built by starting ground-up with some simple concepts (like in the engineer's example) and then being able to retain these concepts through a deep learning architecture?

I don't particularly agree with the view that surrogates are not useful and add an additional layer of opacity. For example, simple, let's say vector autoregressive models can still allow us to detect some interesting relationships in the data that would in all likelihood be retained by  the deep time series. One should not minimize the value of input-level features explainability and its importance, e.g., in finance, or anywhere where the inputs are meaningful categorical variables. Can the authors comment on this?

**Alternative Views Section:**

Yes

**Compliance With Llm Reviewing Policy A Conservative:**

Affirmed.

**Discussion Potential:**

3

**Ethical Review Concerns:**

None.

**Final Justification:**

My concerns have been acknowledged and addressed in the rebuttal.

**Paper Summary:**

This paper focuses on the question of interpretability of deep learning models for time series data. The paper makes the point that there is a fundamental misalignment between current approaches to the explainability of time series models and the way that experts who are familiar with the problem reason about it. This is known as "semantic alignment". They give some concrete examples of the disconnect between these two approaches as they may be seen from an expert's point of view.

The paper then introduces a formal definition of SA for time series models and a proposition to construct interpretable deep time series models. The main thrust of the proposed approaches has to do with explicitly incorporating objectives into the model that allow tracking and alignment of the model's hidden states with some concrete human-understandable concepts. Some potential concrete approaches to this problem are provided.

**Position:**

Yes

**Position In Title:**

Yes

**Related Work:**

2

**Strengths And Weaknesses:**

A major strength of this paper is that it clearly formulates a problem that is arguably at the very core of interpretable time series models. This is a problem that, if solved, would provide a major advance into the use of these powerful models in settings where the ability to be interpretable is key. The proposed approaches are sensible and provide a good roadmap to follow.

One weakness of the paper is that it is somewhat general in its prescriptions. It is hard to disagree with the majority of its suggestions, but the roadmap should be a bit more complete given the relative maturity of the field, at least from an application point of view. The authors acknowledge the speculative nature of some of their proposals, e.g., the objective function in section 4.3. I think that the end result of reading this paper should not be simply to agree with the authors' position, which is hardly controversial, but to allow the reader to start thinking concretely about how to make their (simple) deep learning based time series models interpretable.

**Support:**

2

---

> ### Author Rebuttal · Authors · 2026-03-30
>
> We thank the reviewer for their feedback. We address the main points below.
>
> ---
>
> **W1 - More concrete guidelines**
>
> > One weakness of the paper is that it is somewhat general in its prescriptions. [...] I think that the end result of reading this paper should [...] allow the reader to start thinking concretely about how to make their (simple) deep learning based time series models interpretable.
>
> We fully agree with this objective. This is the motivation behind our formal problem setting (**Sec. 2.1**), definitions (**Sec. 3.2**), and concrete speculations (loss and architecture components in **Sec. 4**), which aim to move away from high-level intuitions (as the reviewer acknowledges).
>
> While this is more prominent in the first sections, we acknowledge that the later sections would benefit from more concrete guidelines. This was a common concern among reviewers, so please allow us to point to other responses that collectively we hope would answer your concern.
> Specifically, in the revision, we will integrate the following three aspects into the main text:
>
> - a **section providing practical guidelines** to follow when designing an interpretable deep learning architecture for time series (please see response **W1/Q1 to Reviewer 1j78**);
> - **an overview figure** of these guidelines for visual support (please see https://anonymous.4open.science/r/rebuttal_figures-D5D4/figures.png);
> - a concrete **instantiation** of the proposed blueprint for the case of system monitoring (please see response **W3/Q3 to Reviewer wZGK**).
>
> We hope this resolves the concern by complementing the blueprint with concrete guidelines to follow.
>
> ---
>
> **Q1 - How to build an interpretable model**
>
> > can the authors argue that a deep learning model can be built by starting ground-up with some simple concepts [...] and then being able to retain these concepts through a deep learning architecture?
>
> Exactly, this is the approach we advocate for in the paper. More precisely, the first step would be to think about which concepts humans would use to reason about the target phenomenon. Then, a neural network can be designed to have such **concepts variables as some of the internal nodes** so that they could directly be part of the model's reasoning [1] (**Sec. 4.1**). This makes the model interpretable and permits human interventions. These are the basics of the concept-based interpretability paradigm, of which in **Sec. 4.2** we hypothesize a concrete extension to the temporal domain.
>
> Note that forcing the network to reason through such interpretable concepts is likely to reduce the network expressive power. Strategies to mitigate this are discussed in **Obs. 4.3**, **Sec. 6.3** and in the newly added guidelines (response **W1/Q1 to Reviewer 1j78**). How to retain these concepts while preserving expressivity in a temporal setting is one of the open questions we posit in the paper.
>
> ---
>
> **Q2 - Surrogate models**
>
> > I don't particularly agree with the view that surrogates are not useful and add an additional layer of opacity. For example, [...] vector autoregressive models can still allow us to detect some interesting relationships in the data that would in all likelihood be retained by the deep time series.
>
> We agree on the importance of explanation methods. In general, they can provide qualitative insights into model behavior, support hypothesis generation, and help identify spurious correlations or dataset biases, especially in exploratory settings.
>
> However, surrogate / post-hoc / explanation methods **cannot provide formal guarantees** on whether the extracted patterns are what is actually used by the deep model inference. The intuition is that the explanation model is a *different* model than the one used for predictions (note how this is not related to how interpretable the input space is). In support of this, there is established evidence both empirical [2] and theoretical [3]. The theoretical result holds irrespective of the input domain, and therefore also applies to time series data.
>
> A meeting point between explanations and interpretability-by-design is that explanations could, for example, be used for discovering patterns that might be relevant for concept discovery.
>
> We will revise the text to highlight the complementary advantages of explanations.
>
> ---
>
> We hope to have clarified all of the reviewer’s concerns and are happy to provide further details if needed.
>
> ---
>
> **Refs:**
> 1. Koh et al., Concept bottleneck models, ICML, 2020.
> 2. Adebayo et al., Sanity checks for saliency maps, NeurIPS, 2018.
> 3. Bilodeau et al., Impossibility theorems for feature attribution, PNAS, 2024.

---

> > ### Author Rebuttal · Reviewer_hdEe · 2026-04-01
> >
> > Thank you for the detailed response!

---

### Decision · Program_Chairs · 2026-04-30

**Decision:**

Accept (regular)

**Comment:**

The reviewers acknowledge that the paper presents a clear and well-motivated argument for semantic alignment (SA) as a core requirement for interpretability in deep time series models.

While initial concerns focused on the generality of the framework and limited implementation guidance, the rebuttal satisfactorily clarified key ambiguities, particularly around mechanisms, concept construction, and feasibility.

The work is best understood as a position paper: not proposing a fully specified method, but offering a rigorous conceptual foundation and research agenda. Given its strong problem formulation, coherent framework, and persuasive responses to critiques, the reviewers accept the paper’s central argument and its contribution to advancing interpretable modeling.